# Nuclear Sirtuins and the Aging of the Immune System

**DOI:** 10.3390/genes12121856

**Published:** 2021-11-23

**Authors:** Andrés Gámez-García, Berta N. Vazquez

**Affiliations:** 1Chromatin Biology Laboratory, Josep Carreras Leukaemia Research Institute (IJC), Ctra de Can Ruti, Camí de les Escoles s/n, 08916 Badalona, Spain; agamez@carrerasresearch.org; 2Unitat de Citologia i d’Histologia, Departament de Biologia Cellular, Fisiologia i Immunologia, Universitat Autònoma de Barcelona (UAB), Cerdanyola del Valles, 08193 Barcelona, Spain

**Keywords:** sirtuins, epigenetics, aging, immune system, immune senescence, inflammation

## Abstract

The immune system undergoes major changes with age that result in altered immune populations, persistent inflammation, and a reduced ability to mount effective immune responses against pathogens and cancer cells. Aging-associated changes in the immune system are connected to other age-related diseases, suggesting that immune system rejuvenation may provide a feasible route to improving overall health in the elderly. The Sir2 family of proteins, also called sirtuins, have been broadly implicated in genome homeostasis, cellular metabolism, and aging. Sirtuins are key responders to cellular and environmental stress and, in the case of the nuclear sirtuins, they do so by directing responses to chromatin that include gene expression regulation, retrotransposon repression, enhanced DNA damage repair, and faithful chromosome segregation. In the immune system, sirtuins instruct cellular differentiation from hematopoietic precursors and promote leukocyte polarization and activation. In hematopoietic stem cells, sirtuins safeguard quiescence and stemness to prevent cellular exhaustion. Regulation of cytokine production, which, in many cases, requires NF-κB regulation, is the best-characterized mechanism by which sirtuins control innate immune reactivity. In adaptive immunity, sirtuins promote T cell subset differentiation by controlling master regulators, thereby ensuring an optimal balance of helper (Th) T cell-dependent responses. Sirtuins are very important for immune regulation, but the means by which they regulate immunosenescence are not well understood. This review provides an integrative overview of the changes associated with immune system aging and its potential relationship with the roles of nuclear sirtuins in immune cells and overall organismal aging. Given the anti-aging properties of sirtuins, understanding how they contribute to immune responses is of vital importance and may help us develop novel strategies to improve immune performance in the aging organism.

## 1. Introduction

Cellular and organismal functions inevitably become compromised with age. With the increasing average age of societies, there has been a growth in the research effort being invested in the molecular and cellular basis of aging. The ultimate aim is to extend the lifespan or the health-span through therapies or habits that either delay the advent of degenerative syndromes or palliate their consequences. In 2013, Lopez-Otin et al. defined nine hallmarks of mammalian aging from a cellular perspective. These included alterations in the epigenome, genomic instability, mitochondrial and stem cell dysfunction, and loss of proteostasis [1]. Loss of cellular fitness resulting from the progressive development of these pathological defects eventually impairs organismal functions and is expected to be therapeutically targetable.

In this context, the immune system is now recognized as an important driver of the aging process. For instance, specific deletion of the DNA repair factor Excision Repair Cross-Complementation Group 1 (ERCC1) in hematopoietic stem cells results not only in greater DNA damage in immune cell subtypes but also in premature senescence and organismal aging. Furthermore, transplantation of aged splenocytes into young mice is enough to accelerate the aging phenotype, while transplantation of young splenocytes into old mice reduces senescence markers in several non-immune tissues [2].

Despite this intriguing recent evidence, the mechanisms by which aging affects immune function has long been a matter of interest. Proper immune function largely depends on the concerted action of its diverse components, and immune cell aging is observed across virtually all immune cell types. Therefore, cumulative perturbations in the physiology of the immune system eventually reduce its ability to respond to both exogenous and endogenous insults [3]. The practical consequences of immunosenescence involve defective clearance of damaged and potentially harmful cells; a concomitant increase of senescence markers in non-immune organs; a higher risk of developing cancer, diabetes, neurodegenerative disorders, autoimmunity, and other maladies; and a poor response to infections and vaccines [4,5,6]. One major manifestation of innate immune aging is inflammaging, a persistent low-grade inflammation that gradually leads to, among others, hematopoietic stem cell (HSC) and T cell exhaustion, thus impairing immune function. In addition, inflammation is thought to facilitate the onset of age-dependent diseases [7]. For instance, exacerbated reactivity of microglia (a macrophage type in the brain) has been linked to neurotoxicity and neurodegenerative diseases [8]. Together, these defects in immune function hugely undermine organismal physiology, highlighting the central role of immunosenescence in the overall aging process.

Sirtuins are an evolutionarily conserved family of proteins that harbor NAD^+^-dependent deacetylase and ADP-ribosyltransferase enzymatic activities [9]. In mammals, there are seven sirtuins [10], which can be localized either in the nucleus, as is the case for SIRT1, SIRT6, and SIRT7, or in the mitochondria, as is the case of SIRT3, SIRT4, and SIRT5. SIRT2 is mostly found in the cytoplasm but binds to chromosomes during mitosis. Sirtuins promote cellular adaptation to stress by regulating epigenetics in the nucleus, cellular metabolism in the mitochondria, and the crosstalk between them. Nuclear sirtuins regulate chromatin function through complex regulatory networks of histone and nonhistone substrates. In this regard, sirtuin-dependent regulation of epigenetic information is closely associated with lysine (K) acetylation (ac) of histones H3 and H4, including H4K16ac, H3K9ac, H3K56ac, H3K18ac, and H3K36ac (Figure 1). Importantly, sirtuin-dependent histone deacetylation is tightly linked to the regulation of histone methylation (me) of the same or nearby lysine residues. For instance, SIRT1 orchestrates heterochromatin formation through H3K9ac deacetylation and suppressor of variegation 3-9 homolog 1 (SUV39H1) activation [11], thereby increasing H3K9me3, an archetypal heterochromatic mark. During G2/M transition, SIRT2 supports chromosome condensation by deacetylating H4K16ac and concomitantly activating PR/SET domain-containing protein 7 (PR-SET7) methyltransferase to induce monomethylation of H4K20 (H4K20me1) [12], another epigenetic mark with repressive functions. SIRT6 plays important roles in gene silencing [13], DNA damage repair [14], and chromosome segregation [15] through H3K9ac, H3K56ac, and H3K18ac regulation, respectively, and SIRT7 functions in gene [16] and retrotransposon silencing [17] and DNA damage repair [18] through H3K18ac and H3K36ac deacetylation. It is of particular note that, under cellular stress, SIRT7 auto-ribosylation leads to its recruitment to chromatin through macro-H2A1 interaction, resulting in gene transcriptional adaptation [16].

Sirtuins are ubiquitously expressed proteins with important roles in numerous tissues [19]. At the organismal level, nuclear sirtuins play major roles in determining the onset of aging and the health-span. SIRT1, SIRT6, and SIRT7 are associated with human and rodent longevity [20,21,22], and mice overexpressing SIRT6 throughout the body or SIRT1 specifically in the brain both prolong the lifespan [23,24]. Accordingly, mice deficient for *Sirt1*, *Sirt6* and *Sirt7* genes develop progeroid-like syndromes [18,25,26], and *Sirt2*-deficient mice have an increased cancer risk [12,27]. Sirtuins are highly expressed in immune cells and play multiple roles in cytokine production, inflammation, and the development of innate and adaptive responses. Here, we review the role of nuclear sirtuins in immune cells and discuss their connections with the aging of the immune system.

## 2. HSCs

HSCs are responsible for long-term generation of blood cell types and can be classified as long- and short-term HSCs. Long-term HSCs (LT-HSCs) are a quiescent population that sustain life-long generation of blood cell types. They first differentiate into short-term HSCs (ST-HSCs), which are able to reconstitute the myeloid and lymphoid compartment for several weeks (Figure 2). During aging, HSCs undergo substantial age-related deterioration. Cell-intrinsic and cell-extrinsic alterations progressively prime aged HSCs to form a phenotype that resembles that of activated young HSCs [28]. At steady state, young HSCs are quiescent unless organismal challenges, particularly infections, arise, in which case they dramatically switch their metabolism to sustain the massive de novo production of immune cell progenitors. Thus, the increased frequency of these challenges with age, together with chronic inflammation and organismal stressors, gradually disrupt HSC resting. Abnormal HSC activity eventually gives rise to HSC exhaustion, reduced regeneration capacity and myeloid bias [29]. Furthermore, clonal hematopoiesis helps hierarchically dampen the fitness of the immune system as it ages. In aged HSCs, defective DNA repair, cumulative DNA damage, and replication stress increasingly cause genomic instability that can be clonally inherited by their cellular progeny [30]. Thus, HSC aging affects the quantity and quality of progenitor and mature immune cells.

Concomitantly, metabolic deregulation, epigenetic alterations, and loss of mitochondrial homeostasis are key hallmarks of HSC aging [28,29]. These defects are highly interconnected, and sirtuins have been proposed to be situated at their crossroads [29]. Indeed, *SIRT1*, *SIRT2*, and *SIRT7* are downregulated during aging in HSCs (Figure 2 and Figure 3), and *SIRT7* expression is reduced in senescent iPSCs [31,32,33]. Further, the SIRT1 and SIRT2 target histone mark H4K16ac is reduced in aged HSCs [34].

Several studies have reported SIRT1 to be essential for HSC integrity and for maintaining their self-renewal capacity and lineage specification. *Sirt1^-/-^* HSCs recapitulate several characteristics of aged HSCs [39]. Similar to what is observed during aging, *Sirt1^-/-^* HSCs escape quiescence and exhibit increased DNA damage and ROS accumulation. Notably, the activity of the transcription factor Forkhead Box (FOXO3), which sustains quiescence and self-renewal capacity in HSCs, is positively regulated by SIRT1 deacetylation in HSCs and other cell types. SIRT1 deletion in adult mice renders HSCs myeloid-biased and induces anemia and lymphopenia. Likewise, several genes commonly upregulated in aged HSCs show increased expression upon *Sirt1* deletion. During aging, the number of HSCs paradoxically increases as a consequence of the loss of quiescence, which ends up reducing HSC regenerative capacity. Accordingly, in the absence of SIRT1, the frequency of LSK (Linage-Sca-1+Kit+, a heterogeneous cellular population containing HSCs) cells and LT-HSCs increases, although the frequency of ST-HSCs is unaffected [39]. In contrast, acute pharmacological inhibition of SIRT1 with Sirtinol (Table 1) in murine fetal LSK cells reduces the frequency of LSK cells, indicating that temporal or chronic loss of SIRT1 activity can have different repercussions on HSC biology. In ex vivo-cultured LSK cells, the pan-sirtuin inhibitor nicotinamide (NAM) promotes HSC differentiation, while the sirtuin agonist resveratrol sustains stemness by repressing HSC differentiation. Similarly, in vitro-cultured LSK cells from *Sirt1^-/-^* mice show lower self-renewal capacity as a consequence of a mechanism involving FOXO suppression, p53 activation, and ROS accumulation [35].

**Table 1 genes-12-01856-t001:** Summary of sirtuin-targeting compounds and their effects in immune cell populations.

Compound	Function	Role in Immune Cells	Species Described
Resveratrol	Sirtuin activator	Increases LSK frequency ex vivo [35]	Mouse
		Limits T cell proliferation in vivo and ex vivo [40]	Mouse
		Enhances SIRT1 deacetylase activity towards c-Jun in T cells [40]	Mouse
		Reduces Th17 differentiation ex vivo and in vivo [41]	Human and mouse
		Enhances SIRT1 deacetylase activity on STAT3 in Th17 cells [41]	Human and mouse
Nicotinamide	Sirtuin inhibitor	Decreases LSK frequency ex vivo [35]	Mouse
		Promotes LSK differentiation into granulocytes ex vivo [35]	Mouse
		Limits macrophage self-renewal through cell cycle control [42]	Mouse
		Increases Treg frequencies and activity ex vivo [43,44]	Human and mouse
		Increases FOXP3 acetylation and protein stability in Treg cells [43]	Human and mouse
Metformin & SRT-1720	SIRT1 agonists	Reduces Th17 differentiation ex vivo and in vivo [45]	Human and mouse
		Enhances SIRT1 deacetylase activity on STAT3 in Th17 cells [41]	Human and mouse
		Reduces AID expression and impairs CSR in B cells (SRT-1720 only) [46]	Mouse
Sirtinol	SIRT1 antagonist	Reduces LSK and LT-HSC frequency and numbers in vivo [35]	Mouse
		Diminishes HSC repopulation capacity [39]	Mouse
Cambinol	SIRT1/2 inhibitor	Promotes a proinflammatory phenotype in macrophages ex vivo [47]	Human
S6	SIRT6 inhibitor	Limits the differentiation of monocytes to dendritic cells ex vivo [48]	Human

Although SIRT1 downregulation has been reported in aged HSCs, there are some conflicting findings in this regard, so it is a matter of debate. Rimmelé and coworkers reported in mice that *SIRT1* expression was higher in LSK cells than in total bone marrow (BM) cells, while aged murine HSCs expressed reduced *SIRT1* levels [39]. In contrast, Chambers et al. did not find any age-related transcriptional downregulation of *SIRT1* in the murine HSCs [32]. The study conducted by Xu et al. did not reveal differential expression of *SIRT1* in LSK cells from aged mice, either. However, they reported an interesting mechanism by which SIRT1 protein levels are post-transcriptionally decreased due to selective autophagic degradation of the SIRT1 protein [49].

Sirtuins have also been implicated in the preservation of mitochondrial integrity in HSCs during aging. Indeed, SIRT2 has been linked to the maintenance of HSC homeostasis in aged mice via suppression of the NLR family pyrin domain containing 3 (NLRP3) inflammasome, a multimeric protein complex involved in sensing damage- and pathogen-associated molecular patterns. In aged HSCs, mitochondrial stress can trigger the activation of NLRP3, and aberrant activation of the NLRP3 inflammasome is known to drive cell death and functional decline in HSCs during aging. While young *Sirt2^-/-^* mice have normal HSC frequencies with full regenerative capacity, that of HSCs is lower in old *Sirt2^-/-^* mice. Notably, *SIRT2* is expressed at reduced levels in aged HSCs, which is associated with greater NLRP3 inflammasome activation, thereby representing a plausible mechanism of HSC decline in aged *Sirt2^-/-^* mice [36].

SIRT6 has not been directly related to HSCs in the context of aging, but it is known to be necessary for HSC quiescence and welfare. SIRT6 deficiency results in a progeroid HSC phenotype due to epigenetic dysregulation of Wnt signaling. HSC homeostasis is largely influenced by Wnt ligands and signaling, which help maintain HSC integrity [50,51]. SIRT6 interacts with the Wnt signaling transcription factor LEF1 (lymphoid enhancer binding factor 1), which recruits SIRT6 to Wnt target genes. At the promoters of these genes, SIRT6 deacetylates its target H3K56ac and thereby silences their expression. In the absence of SIRT6, Wnt target genes become overexpressed, resulting in aberrant HSC proliferation that ultimately leads to HSC exhaustion and diminished self-renewal capacity. This *Sirt6* deficient phenotype can be reversed by inhibition of Wnt signaling [37].

SIRT7 is highly expressed throughout the hematopoietic system [52]. SIRT7 knockout mice exhibit a whole-body progeroid phenotype [18], and *Sirt7^-/-^* HSCs show several characteristics of aged HSCs: loss of quiescence, myeloid bias, and increased propensity to enter the cell cycle when stimulated with cytokines ex vivo [31]. In HSCs, SIRT7 represses the expression of several genes coding for mitochondrial ribosomal proteins and transcription factors by direct interaction with their promoters in a nuclear respiratory factor 1 (NRF1)-dependent manner. Aged immune cells exhibit mitochondrial dysregulation, which involves inefficient mitochondrial function and increased mitochondrial mass [38]. Mitochondrial dysfunction leads to the accumulation of misfolded proteins and invokes the mitochondrial unfolded protein response (mtUPR), an adaptive reaction that aims to recover the compromised mitochondrial proteostasis. *Sirt7^-/-^* HSCs display increased mitochondrial mass and enhanced basal expression of mtUPR genes, while *SIRT7*-knocked-down (KD) cells show inefficient clearance of misfolded proteins. This suggests that *Sirt7^-/-^* HSCs are subject to constitutive mitochondrial stress, which makes them adopt an aged immune cell phenotype. In aged HSCs, ribosomal DNA (rDNA) transcription has been linked to replication and increased DNA damage. SIRT7 is a major regulator of rDNA transcription [52], acting through the control of different components of the basal transcriptional machinery, but whether this function may play a role in proteostasis and metabolism in aged HSCs is not known.

## 3. Innate Immunity

Innate immunity involves a variety of cell types, including natural killer (NK) cells, various macrophage populations, monocytes, dendritic cells, neutrophils, eosinophils, and basophils (Figure 2). Innate immune cells originate from hematopoietic stem cells in the bone marrow and, in some cases, through direct self-renewal [53]. In adulthood, innate immune cells reside in most of our tissues, where they play important roles in responding to external threats and in tissue homeostasis. The abundance, distribution and function of innate immune cells are markedly altered with aging, the long-term and constitutive low-grade secretion of proinflammatory cytokines being an important feature contributing to the aging process. Indeed, this state of chronic inflammation, or inflammaging, is a typical feature of immunosenescence that actively contributes to the deterioration of immune and non-immune tissues [54]. Sirtuins regulate the function of innate immune cells at multiple levels with important implications for immune and organismal aging (Figure 2). For the most part, sirtuin downregulation in human innate immune cells is associated with proinflammatory processes, and their overexpression is thought to protect tissues. Similarly, whole-body or myeloid-specific sirtuin deficiency in mice results in the development of different inflammatory conditions, including autoimmunity, obesity, and neurodegeneration [55,56,57,58].

## 4. Monocytes and Macrophages

Macrophages reside in all tissues of the body [59]. They are critical for tissue homeostasis during the development of context-specific functions, as is the case of alveolar macrophages in the lung or microglial cells in the central nervous system.

In addition to their tissue-specific roles, macrophages are also well known for their ability to phagocyte pathogens, which leads to antigen presentation and inflammation. Macrophages originate from erythro-myeloid precursors during early embryo development, or from infiltrating monocytes in adulthood. Embryo-derived and monocyte-derived macrophages are both capable of maintaining their abundance through self-renewal when required. Macrophages respond to the environment, resulting in the acquisition of a spectrum of functional states. Upon antigen stimulation, macrophages activate and polarize into a pro- or anti-inflammatory phenotype, the so-called classically activated M1 and alternative M2 macrophages, respectively. M1 macrophages perform cytotoxic and tissue-damage proinflammatory functions, while M2 macrophages are important for resolving inflammation and tissue repair. Macrophage function is altered in aged hosts, resulting in poor outcomes after infections and tissue degeneration [60]. The phenotypic features of aged macrophages may differ depending on the macrophage population, but many studies indicate that macrophages from old hosts have impaired phagocytic capacity and are skewed towards a more pro-inflammatory phenotype. Macrophage depletion in aged mice under immunotherapy administration is associated with reduced proinflammatory cytokine production and survival [61]. Similarly, macrophage targeting in aged mice improves peripheral nerve structure and muscular performance [62]. Together this data indicates that macrophage deregulation with age is a major contributor to overall organismal aging.

Various lines of evidence indicate that nuclear sirtuins support immunosuppressive functions and M2-associated responses (Figure 2 and Figure 4). For instance, SIRT6 expression increases in mouse BM macrophages under M2-polarizing conditions [55]. Similarly, SIRT2 expression decreases in mouse microglia upon LPS stimulation, which induces M1 polarization [56]. Sirtuins support macrophage biology at many levels, including its cellular differentiation, self-renewal, polarization, and activation. SIRT1 and SIRT2 protein levels increase during the differentiation of human monocytes to macrophages, and their inhibition with cambinol (Table 1) or deficiency prompts the development of a proinflammatory phenotype [46]. SIRT1 and SIRT2 prevent the premature expression of proinflammatory genes through the control of their chromatin structure. Mechanistically, SIRT1 and SIRT2 interact with DNA methyltransferase 3B(DNMT3B) enzyme to promote DNA methylation in addition to limiting H3K4me3 and H3K27ac deposition [47]. BM-derived macrophages from *Sirt6^f/f^ LysM-Cre* mice, in which the SIRT6 gene is specifically deleted in myeloid cells, have increased levels of expression of proinflammatory cytokines, including interleukin (IL)-6, tumor necrosis factor α (TNF-α), and interferon β (IFN-β), and increased migration capabilities compared to WT controls, but it is not known whether this is due to impaired cellular differentiation [55].

SIRT1, SIRT2, SIRT6 and SIRT7 activities are also important for the balance of macrophage polarization (Figure 4), which depends on specific stimuli and downstream signaling events [66]. M1 polarization happens in response to triggers such as LPS or IFN-γ and strongly depends on the nuclear factor kB (NF-κB) transcription factor, a master regulator of inflammation and age-related pathways [13,66]. M2 polarization is induced by stimuli such as IL-4 or IL-10 and uses different cascade signals, including signal transducer and activator of transcription 6 (STAT6) and peroxisome proliferator-activated receptor γ (PPARγ) activation. Many studies have shown that SIRT1, SIRT2 and SIRT6 limit macrophage inflammation through NF-kB regulation [55,57,58]. SIRT1, SIRT2, and SIRT6 knockout BM macrophages display hyperacetylation of the NF-kB p65 subunit, which raises its transcriptional activity, and increased expression of NF-kB target genes, including IL-6, TNF-α, and IL-1β. The biochemical and functional interplay of SIRT1, SIRT2, and SIRT6 with NF-kB is well documented in many cell types [13,65,67], suggesting that similar mechanisms of NF-kB regulation may exist in macrophages. In HeLa cells, SIRT6 silences the expression of NF-kB targets genes by deacetylating H3K9ac [13]. In addition, in 293F cells, SIRT6 promotes the expression of the NF-κB repressor, IκBα (nuclear factor of kappa light polypeptide gene enhancer in B-cells inhibitor, α), via a mechanism that involves cysteine monoubiquitination of the histone methyltransferase SUV39H1, which results in its dissociation from the IκBα gene promoter and, consequently, gene activation [68]. In mouse embryonic fibroblasts, SIRT2 directly deacetylates the p65 subunit of NF-kB at lysine 310, repressing its transcriptional activity [67]. SIRT2 deacetylates H4K16ac during the G2/M transition, but whether SIRT2 epigenetically regulates NF-kB targets genes during inflammation or under similar conditions has not been explored. Finally, *SIRT7* expression has been reported to decrease in an age-dependent manner in leukocytes from healthy patients. In the monocytic THP-1 cell line, PMA-mediated monocyte-to-macrophage differentiation increases *SIRT7* expression, while SIRT7 overexpression increases differentiation markers in non-stimulated THP-1 cells [64].

SIRT1 and SIRT2 also play important roles in microglia activation and brain inflammation, which have significant implications for age-dependent neurodegenerative diseases [56,69]. SIRT1 overexpression in microglial cells also protects neural cells from amyloid-β peptide-induced death, a neurotoxic pathway related to the pathogenesis of Alzheimer disease [69]. *Sirt2^-/-^* mice and *SIRT2* KD microglial cells challenged with LPS have a stronger microglial proinflammatory response, including higher levels of cytokine secretion and free-radical production, and cell death [56]. At the molecular level, SIRT1 and SIRT2 exert their anti-inflammatory properties by downregulating NF-kB activity. SIRT2 enzymatic capabilities are modulated by phosphorylation, and the absence of this posttranslational modification on serine S331 of SIRT2 prevents NF-kB acetylation in microglial cells. Indeed, overexpression of the phospho-resistant SIRT2 S331A mutant, but not of the phospho-mimetic SIRT2 S331D mutant, in microglia results in diminished p65 subunit acetylation at lysine 310, possibly bringing about NF-kB target-gene silencing.

Although macrophages are terminally differentiated cells, they have the capacity to self-maintain through local proliferation independently of hematopoietic precursor differentiation, a feature normally associated with stem cells [70]. SIRT1 participates in macrophage self-renewal by controlling cell-cycle progression and proliferation [42]. *SIRT1* KD macrophages are less efficient in colony formation assays and display a G1 cell cycle arrest that is associated with Myc downregulation, impaired phosphorylation of E2 factor (E2F), and increased nuclear translocation of FOXO1 transcription factor. Accordingly, SIRT1 deficiency results in gene silencing of the Myc and E2F pathways, which play important roles in self-renewal, and upregulation of those pathways involving FOXO factors, which are known to induce cell-cycle arrest. A similar phenotype is observed in macrophages treated with NAM (Table 1), which raises the possibility that other sirtuins are also involved in the self-renewal process.

Despite the well-established anti-inflammatory roles of sirtuins, only one study has addressed their contribution to macrophage aging and the development of age-related diseases. The presence of senescent cells in aged tissues promotes M1 macrophage polarization and activation, resulting in tissue inflammation and compromised insulin signaling [71]. Indeed, chronic low-grade inflammation in the elderly is associated with insulin resistance and diabetes [72]. In this regard, myeloid SIRT2 has been shown to protect against glucose intolerance by controlling aged-related inflammation [63]. How SIRT2 regulates this process is explained by its functional interplay with the NLRP3 inflammasome (Figure 4), as also reported in HSCs. In macrophages, SIRT2 interacts and deacetylates the NLRP3 scaffold protein to suppress NLRP3 inflammasome assembly and activity. Importantly, SIRT2 levels decrease with age in macrophages in conjunction with increased NLPR3 acetylation and activation. In addition, white adipose tissue previously co-cultured with aged macrophages exhibits impaired insulin signaling compared with young controls, which can be rescued with old macrophages transduced with SIRT2 or with a constitutive deacetylated NLPR3 form. This study highlights the SIRT2-NLPR3 axis in macrophages as an interesting target for reversing age-associated inflammation and improving glucose homeostasis.

## 5. Eosinophils

Eosinophils play important roles in defending against helminth parasite infections and allergic inflammation, such as allergic rhinitis and asthma. Other roles include cellular metabolism, thermogenesis, and antitumoral responses. Eosinophils are produced in the bone marrow in the presence of IL-5, a process that critically depends on GATA-1 transcription factor [73]. In humans and mice, recent evidence indicates that eosinophil frequencies diminish in the white adipose tissue of aged hosts [74]. This age-dependent drop in eosinophil abundance is correlated with the occurrence of inflammaging and the development of different age-related conditions, including frailty and impaired immune response to immunization. Importantly, transfer of young eosinophils into aged mouse recipients reduces systemic low-grade inflammation, improves physical performance, and immune differentiation and activity, highlighting the role of young eosinophils as rejuvenating agents.

Our knowledge of the role of sirtuins and eosinophils is very limited (Figure 2 and Figure 5). There is only one report describing the functional interplay between them (Figure 5A) [75]. However, some evidence suggests that sirtuins play important roles in eosinophil biology. For instance, the DNA damage response is a process that is strongly connected to nuclear sirtuin activity [76], and it is known to be more robust in eosinophils than in other innate immune cells [77]. SIRT6 is important for eosinophil differentiation and function [75]. In vitro differentiation of BM cells into eosinophils is altered in the absence of SIRT6. In addition, eosinophil-mediated M2 macrophage polarization, a process that depends on eosinophil IL-4 secretion, is also impaired in the presence of *Sirt6^-/-^* eosinophils. SIRT6 regulates the abundance and activity of GATA-1, a transcription factor required for eosinophil lineage commitment and differentiation [73]. Intriguingly, SIRT6 promotes GATA-1 transcriptional activity independent of its enzymatic activity. SIRT6 forms a ternary complex with GATA-1 and p300 to positively regulate GATA-1 activity, so it is possible that SIRT6 acts as a scaffold protein to recruit p300 acetyltransferase to this complex. Similar to what happens in aged hosts, after exposure to the cold, myeloid *Sirt6^-/-^* mice have lower frequencies of eosinophils in the white adipose tissue than do WT mice. Adaptive thermogenesis requires production of cytokines, including IL-4 by eosinophils, which leads to M2 macrophage polarization and in turn facilitates the browning of white adipocytes and heat generation. Although this study addressed the role of eosinophil SIRT6 in brown adipocyte activity, it is of note that brown adipose tissue depots and function decline in the elderly [78], suggesting novel avenues for understanding the mechanisms of organismal aging and their potential relationship with eosinophil SIRT6.

## 6. NK Cells

NK cells are cytotoxic lymphocytes with important roles in innate immunity against viral-infected cells and tumors. NK cells secrete perforins and granzymes and express cell-death ligands on their surface to induce target cell apoptosis. In addition, NK cells secrete a variety of pro-inflammatory cytokines, including TNF-α and IFN-β, that have important roles in sustaining and amplifying immune responses through macrophage and dendritic cell activation. In aged humans, the NK cell compartment is associated with an increase in mature long-lived circulating NK cells [83]. Despite this increase, NK cell-mediated cytotoxicity, including granule secretion and death receptor-mediated killing, is impaired, resulting in poor responses to viruses and an increase in cancer development.

The role of sirtuins in NK cell function has been little investigated (Figure 2 and Figure 5). Levels of human SIRT1 expression are high in aged NK cells [84]. In particular, the level of SIRT1 expression is significantly higher in human NK cells of subjects older than 85 years than in senior and young people with average ages of 75 and 21 years, respectively. Similarly, the levels of heat shock protein 70 (HSP70), a protein with important roles in protein folding and a downstream effector of SIRT1 activity in the control of protein quality [85], are also high in senior people aged over 85 years. In the same group, superoxide dismutase 2 (SOD2), a major antioxidant enzyme regulated by SIRT1 in many cells [86], is also strongly expressed in activated NK cells. Further investigation may help us understand the role of SIRT1 in aged NK cells and show whether SIRT1 has a functional relationship with HSP70 and SOD2.

SIRT2 promotes the activity of liver NK cells in response to hepatocellular carcinoma (Figure 5B) (HCC) [79]. SIRT2 expression specifically increases in liver NK cells from HCC-induced mice, where it promotes NK cell activity. SIRT2-overexpressing NK cells secrete higher proinflammatory cytokines, cytotoxic granules, and have enhanced tumoricidal activity, whereas *SIRT2* KD impairs NK cell cytotoxic activity. SIRT2 activity is correlated with increased phosphorylation of extracellular regulated kinase 1/2 (Erk1/2) and p38, two signaling pathways important for NK cell activity. Despite the importance of SIRT2 to the liver NK cell-mediated anti-tumoral response, the role of this sirtuin in NK cell aging is yet to be explored.

## 7. Dendritic Cells

Dendritic cells (DCs) are antigen-presenting cells that have important roles in adaptive immunity and the maintenance of self-tolerance. At steady state, dendritic cells are highly phagocytic and continuously present self-antigens to limit T cell reactivity. Upon infection, DCs mature, resulting in the increased expression of costimulatory receptors, including CD80, CD86, and MHC-II molecules, secretion of proinflammatory cytokines, and in T cell priming. Major dendritic subtypes include conventional dendritic cells (cDCs), of myeloid origin, and plasmacytoid dendritic cells (pDCs), which originate from a lymphoid precursor. Aging causes major changes in DC activity. In general terms, while DC response to pathogens is diminished, there is an increased reactivity to self-antigens and increased expression of proinflammatory cytokines, which contribute to tolerance breakdown and inflammaging [87].

While SIRT1 is dispensable for DC differentiation and maturation, DC SIRT1 is of great importance in maintaining the balance of Th-mediated immune responses (Figure 2 and Figure 5). Indeed, SIRT1 expression increases in DCs upon Toll-like receptor (TLR) stimulation in humans and mice, and its deletion in mice results in altered T cell polarization [80,81]. However, several research groups have reported contrasting roles for SIRT1 in this cell type (Figure 5C). Yang and colleagues found that DC SIRT1 directs the production of Th17 cells, a T cell subset with inflammatory properties, by restraining the production of IL-27, an anti-inflammatory cytokine that suppresses Th17 differentiation. Indeed, *Sirt1^f/f^ CD11c-Cre* mice, in which SIRT1 is specifically deleted in DCs, have lower percentages of Th17 cells. At the molecular level, SIRT1 interacts with and deacetylates interferon regulatory factor 1 (IRF1), a transcription factor related to IL-27 expression. IL-27 is a protein heterodimer composed of subunits p28 and Epstein-Barr-induced gene 3 (EBI3). SIRT1-dependent IRF1 deacetylation reduces IRF1 binding to the il-27p28 gene promoter, resulting in its silencing, hence leading to reduced IL-27 production and the promotion of Th17 differentiation. Using a similar mouse model of *Sirt1* deletion in DCs, Liu and colleagues reported that DC SIRT1 dictates the balance of Th1 and regulatory T (Treg) cell production upon DC stimulation, with no alteration in the Th17 lineage. Mice with *Sirt1^-/-^* DCs have higher percentages of IFNγ^+^ T cells and IFNγ secretion and lower percentages of FOXP3^+^ T cells and levels of *FOXP3* mRNA. In this study, SIRT1 regulates the production of IL-12 and TGF-β1, two hallmark cytokines for Th1 and Treg differentiation, respectively, in a hypoxia-inducible factor α (HIF1α)-dependent manner. In human DCs, SIRT1 also limits the production of IL-12p70 in response to zymosan, a TLR2 stimulus involved in immunotolerance and Th1 cytokine downregulation [82]. IL-12p70 is a heterodimer of p35 and p40 subunits, which are encoded by the IL-12a and IL-12b genes, respectively. Mechanistically, zymosan prompts SIRT1 recruitment to the IL-12a gene promoter, resulting in chromatin compaction at nucleosome 1 and histone deacetylation, which limits IL-12p35 expression. Overall, these studies indicate that SIRT1 is a major regulator of cytokine production in DCs and has important implications for subsequent T cell subset generation.

In humans and mice, SIRT6 participates in dendritic cell differentiation and maturation (Figure 5D) [48]. Compared with WT controls, *Sirt6^-/-^* mice have fewer cDC precursors in their bone marrow. In addition, in vitro differentiation and maturation of mouse DCs from BM cells are impaired in the absence of SIRT6. More striking results have been obtained in a human model of cDC generation, in which SIRT6 inhibition with S6 inhibitor (Table 1) severely impairs monocyte differentiation into DCs. From a phenotypic perspective, mouse *Sirt6^-/-^* BM derived DCs are less mature, as measured by reduced expression of CD86, CD80, and MHCII, increased endocytic capacity, and a reduced ability to stimulate lymphocyte proliferation. Importantly, TLR engagement with LPS in *Sirt6^-/-^* BM derived DCs results in increased percentages of TNF-α- and IL-6-producing cells, implying that SIRT6 fine-tunes cytokine production in these cells. Overall, this study highlights the important role of SIRT6 in DCs and suggests that lack of SIRT6 in aged DC could be partially responsible for the poor immune responses and inflammaging.

## 8. Adaptive Immunity

While the innate immune system recognizes low-specificity repetitive motifs present in a wide range of pathogens and damaged host cells, the adaptive immune system is remarkable for its high degree of antigen specificity. B and T lymphocytes, the two cellular members of adaptive immunity, are generated in the bone marrow (Figure 2), although T cell progenitors subsequently migrate into the thymus to complete their maturation. Mature B and T cells circulate in the bloodstream and lymph system, and both express B cell receptors (BCRs) or T cell receptors (TCRs) in their membranes that are raised to recognize almost any exogenous or malignant antigens while tolerating self-antigens. Clonal diversity of antigen specificity is thus the cornerstone of the adaptive immune system. Upon recognition of infectious agents, B and T cells are activated and differentiate into effector cells or long-living memory cells. Effector lymphocytes either amplify the innate immune response by specifically targeting pathogens or through cytokine secretion, or induce the death of infected and malignant host cells, or end the immune response once the challenge has been eliminated. In the context of immunosenescence, B and T cell clonal diversity is compromised, and a substantial reduction in the ability to respond to vaccines and new pathogenic agents is observed.

## 9. T Cells

T lymphocytes are subdivided into helper CD4^+^ T and cytotoxic CD8^+^ T cells and are activated by TCR-specific antigens in a process involving cell-to-cell contacts. Cytotoxic CD8^+^ T cells recognize malignant cells or cells infected and target them for cell death by various mechanisms, including the production of granzymes and perforins, two major pro-apoptotic factors. Helper CD4^+^ T cells have immunomodulatory functions and are further divided into myriad subsets, including Th1, Th2, Th9, Th17, and Treg, each of which has a distinctive group of immune cell targets and cytokine expression pattern (Figure 2) [88,89].

Although a certain degree of naïve T cell production is thought to be maintained until old age, it is accepted that the main T cell pool is established early in life. During age-related thymus atrophy, a progressive reduction in thymic cellularity and a marked loss of tissue architecture take place. This is accompanied by an exponential decrease in thymopoiesis, with a half-life of 16 years in humans [90]. The decline in the production of new T cells with age has consequences for the pre-existing naïve T cell pool and for the rest of the immune system.

On one hand, aged naïve T cells become responsible for maintaining the compartment in the absence of a substantial T cell production, which makes them enter a stem-like state [91]. On the other hand, continuous exposure to new pathogens and the appearance of autoimmune disorders and chronic infections result in an enlarged pool of clonally expanded memory T cells at the expense of the peripheral naïve T cell pool. This ultimately limits TCR diversity, dampens the ability of the adaptative immune system to confront new and preexisting challenges, and is a hallmark of immunosenescence [3,92]. T cell aging is accompanied by a series of T cell-intrinsic defects that include T cell exhaustion, extensive genetic and epigenetic alterations, impaired TCR signaling and loss of proteostasis and mitochondrial homeostasis [92]. Sirtuins contribute to T cell biology (Figure 6) and to the preservation of the health-span in the T cell compartment at multiple levels: SIRT1 has a complex role regulating TCR-mediated T cell responses, senescence, and helper T cell polarization; Sirt6 knockout in T cells produces systemic inflammation in mice, and high levels of SIRT7 expression in breast cancer are linked to T cell exhaustion.

Proportional activation of T cells during the immune response depends strictly on a threshold in TCR signaling that determines whether stimulated T cells mount an effective immune response or whether they become anergic. This threshold is crucial to preclude autoimmunity and can become dysregulated during aging [93,94]. TCR signaling is carefully controlled by co-stimulatory or co-repressor receptors at the plasma membrane, and by intracellular modules that fine-tune the intensity of TCR signaling. SIRT1 has emerged as an important factor for adjusting T cell responses by regulating the termination of TCR signaling (Figure 6A and Table 1). In the absence of SIRT1, T cell activation with anti-CD3 antibodies is permitted regardless of CD28 co-stimulation. In mice, TCR-stimulated *Sirt1^-/-^* T cells disproportionally proliferate, produce increased levels of IL-2, and are unable to enter anergy, implying that SIRT1 negatively regulates TCR signaling in vivo [40,95]. In turn, this results in a loss of tolerance in *Sirt1^-/-^* T cells. Indeed, while naïve and activated T cells display similar SIRT1 expression levels, that of anergic T is significantly higher. Mechanistically, SIRT1-mediated termination of TCR signaling involves the transcription factor AP-1, which must be transcriptionally active if T cell effector responses are to be effective [95]. Acetylation of the c-Jun member of the AP-1 heterodimer is needed for it to be active, and SIRT1 dynamically regulates c-Jun acetylation during TCR activation [95,96,97]. Therefore, upon TCR stimulation, when c-Jun acetylation peaks, SIRT1 interacts with c-Jun to reduce its acetylation levels and thereby extinguish the AP-1-mediated TCR response. Providing further evidence that SIRT1 acts as a feedback modulator of T cell activation and anergy, one study demonstrated that IL-2, which can reverse anergy in T cells, suppresses *SIRT1* expression by preventing FOXO3a from binding to the *Sirt1* promoter. This represents a plausible mechanism for recovering TCR sensitivity [98].

**Figure 6 genes-12-01856-f006:**
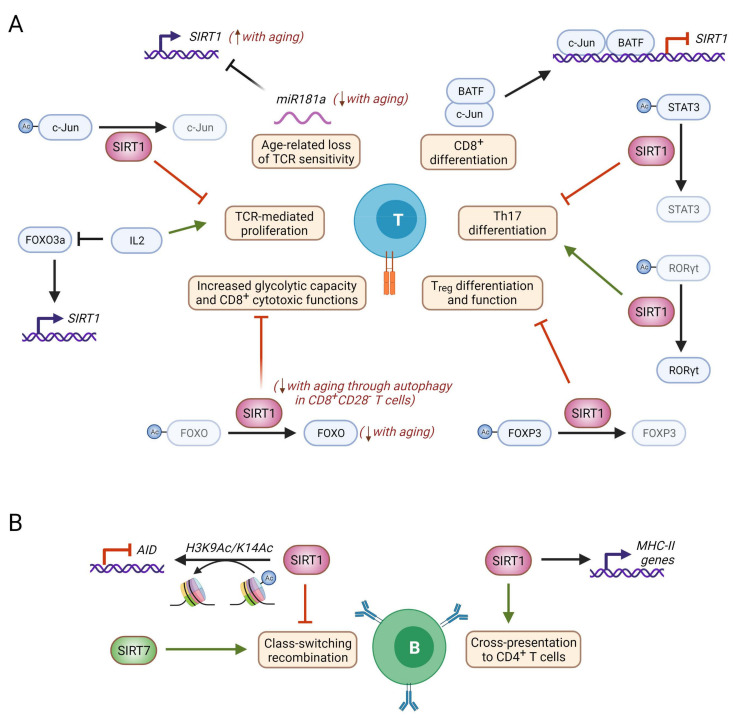
Molecular mechanisms of nuclear sirtuin activity in B and T cells. (**A**) SIRT1 negatively regulates TCR-mediated T cell activation through deacetylation of the AP-1 member c-Jun [40,95,96,97]. Conversely, IL-2 induces exit from anergy by repressing *SIRT1* expression via inhibition of FOXO3a, and BAFT cooperates with c-Jun to repress SIRT1 expression during CD8+ differentiation [98,99]. Age-associated miR181a downregulation in T cells results in SIRT1 upregulation, leading to a loss of TCR sensitivity [100]. SIRT1 balances Th17 differentiation through two opposite mechanisms: on the one hand, SIRT1 deacetylates STAT3, thereby impairing STAT3 nuclear translocation and transcription of the STAT3 target Rorc [41]. Contrarily, SIRT1-mediated deacetylation of RORγt is needed for its proper transcriptional activity. SIRT1 also negatively regulates Treg differentiation and suppressive functions through deacetylation of FOXP3, which enhances its proteasomal degradation [101]. In CD8^+^CD28^-^ T cells, which accumulate during aging, SIRT1 protein levels are dramatically decreased, resulting in a reduction of FOXO1 acetylation and half-life that contributes to increased CD8^+^ cytotoxicity [102,103]. (**B**) In B cells, *SIRT1* deficiency results in reduced levels of MHC-II, which results in impaired cross-presentation to CD4^+^ T cells [104]. SIRT1 is also important for class switching recombination, as it represses AID expression by deacetylation of H3K9Ac and H3K14Ac at the AID promoter [46]. Contrarily, Sirt7^-/-^ splenic B cells display defective class switching recombination [18]. Faded lines indicate age-related loss of function, and comments in red indicate age-related changes. Figure created with BioRender.com.

In aged T cells, different conflicting studies have reported both increased and decreased SIRT1 protein and mRNA levels, indicating the existence of a complex signaling network governing SIRT1 activity in this context (Figure 6A). During CD8^+^ T cell differentiation, IL-12 stimulation increases histone acetylation and the transcription factor basic leucine zipper ATF-like transcription factor (BATF). BATF cooperates with c-Jun to repress transcription of the *SIRT1* gene, ensuring a high level of histone acetylation at the T-bet promoter to drive increased ATP production and T cell differentiation into effector cells [99]. The level of BATF expression is higher in old CD4^+^ T cells, and the accessibility of BATF binding motifs increases as CD8^+^ T cells age [105]. These observations indicate that SIRT1 downregulation may couple T cell activation with T cell aging [106]. In agreement with this model, *Sirt1^-/-^* mice develop spontaneous autoimmunity indicating that SIRT1′s role in the maintenance of peripheral tolerance may be important to prevent this pattern of T immunosenescence [107]. In human elderly individuals, *SIRT1* expression is significantly reduced in peripheral blood mononuclear cells but whether this is due to aberrant epigenetic regulation of the SIRT1 locus and whether it has a significant impact on T cell responsiveness is not known [108].

During T cell aging, downregulation of the microRNA miR-181a in aged naïve and memory T cells has also been described to impact *SIRT1* levels. miR-181a fine-tunes T cell activation by regulating the expression of proteins that affect the intensity and outcome of TCR signaling. In aged human T cells, miR-181a downregulation enhances the expression of several negative feedback regulators of TCR signaling, including SIRT1, thus raising the threshold of T cell activation and reducing T cell sensitivity [100]. Notably, SIRT1 inhibition or silencing in cycling aged human T cells not only restores cell cycle progression but also reduces their replication stress [109]. This is in contrast to observations in primary mouse fibroblasts, in which the absence of SIRT1 is associated with abnormal DNA replication [110].

Overall, these studies indicate that SIRT1 expression is strictly calibrated to ensure proper T cell responses and that SIRT1 deregulation in aged T cells may either predispose to enhanced T responsiveness in the case of SIRT1 downregulation and to poor T cell responses in the case of SIRT1 upregulation.

The accumulation of terminally differentiated CD8^+^CD28^-^ T cells is another hallmark of immunosenescence, and SIRT1 has been linked to the aging of these cells [111,112]. In the absence of CD28 co-stimulation, CD8^+^CD28^-^ T cells are highly cytotoxic, express proinflammatory cytokines and acquire characteristics of replicative senescence [113]. During aging, SIRT1 undergoes autophagy-mediated degradation in multiple murine organs, including the spleen and thymus. In aged CD8^+^CD28^-^ memory T cells, SIRT1 is downregulated at the protein level, with *SIRT1* transcription being unaltered, and inhibition of autophagic degradation restores SIRT1 abundancy [49]. Similar findings were obtained by Jeng and coworkers, who observed that human CD8^+^ memory T cells and, more prominently, terminally differentiated CD8^+^CD28^-^ T cells exhibit dramatically decreased SIRT1 (but not SIRT6 or SIRT7) protein levels, without any alteration in its gene expression. Mechanistically, loss of SIRT1 enhances the proteasomal degradation of its target FOXO1 and thereby increases the glycolytic capacity and the cytotoxic effector functions of these memory T cells (Figure 6A) [102]. Indeed, FOXO1 has recently been reported to prevent senescence and negatively regulate activation and terminal differentiation in CD8^+^ T cells [114]. Therefore, SIRT1 loss during the latest stages of CD8^+^ T cell differentiation could contribute to inflammaging by promoting the accumulation of active and highly cytotoxic CD8^+^ T cells.

In contrast with the anti-inflammatory roles generally attributed to sirtuins, SIRT1 has been shown to contribute to a general pro-inflammatory phenotype by suppressing Treg activity. This is relevant because activated Tregs accumulate at the periphery in aged individuals, probably due to the pro-inflammatory context imposed by age [45,115,116]. FOXP3 deacetylation by SIRT1 makes it more prone to proteasomal degradation, thereby reducing the suppressive Treg function in in vitro suppression assays. Conversely, sirtuin inhibition with NAM significantly increases Treg cell frequency and function in vitro (Table 1) [43,44]. Furthermore, specific deletion of Sirt1 in FOXP3^+^ cells increases FOXP3 levels and Treg function in vivo, thereby improving survival in allograft transplants [118]. As further evidence that SIRT1 promotes pro-inflammatory T cell phenotypes, SIRT1 has been found to be involved in the differentiation of Th17 cells. These are CD4^+^ T cells that have an important inflammatory function in bacterial and fungal infections and that have been linked to several inflammation-associated diseases. SIRT1 is upregulated during Th17 differentiation and deacetylates the central Th17 transcription factor RORγt to optimize its transcriptional activity, so SIRT1 inhibition suppresses both Th17 differentiation and function [101]. Conversely, SIRT1 regulates STAT3 acetylation to determine its cellular distribution. SIRT1 activation with different agonists (Table 1) reduces STAT3 translocation to the nucleus and in turn impairs transcription of the STAT3 target *RORC* (which codes for RORγ), thereby blocking Th17 differentiation [41]. SIRT1 then negatively regulates RORγ levels while increasing its transcriptional activity. Whether these two opposing functions need to be balanced to control Th17 generation and whether this is context-dependent remain to be unexplored. Finally, SIRT1 has also been described as negatively regulating CD4^+^ T cell differentiation into Th9 cells via a mechanistic target of rapamycin (mTOR)–HIF1α-dependent mechanism [103]. Although the role of SIRT1 in effector helper T cells during aging has not yet been investigated, the established importance of SIRT1 in fate decisions during helper T cell differentiation suggests that SIRT1-deregulated expression and activity probably affect the imbalance in CD4^+^ T cell subpopulations that is observed at the onset of aging and disease.

In a paper studying the gene expression changes occurring during immunosenescence in rats, SIRT2 protein levels were found to be significantly decreased in the spleen and, more prominently, in the thymus of aged rats. This was in contrast with the fact that old thymus also displayed reduced levels of the SIRT2 target H4K16Ac [118]. More generally, H4K16Ac hypoacetylation has previously been linked to replicative senescence and found to be relatively weak in several aged murine tissues. The authors proposed a plausible explanation for the lower levels of both SIRT2 and H4K16Ac, wherein the weaker association of MOF, the main H4K16-acetyltransferase, with the nuclear lamina was responsible for the hypoacetylation of H4K16 [119].

Changes in DNA methylation and loss of silent heterochromatin regions during aging, and, in particular, in immunosenescence are the epigenetic disorders most commonly recognized as appearing with advancing age. In this context, the heterochromatin mark H3K9me3 is known to be weaker in aged humans. Although its age-dependent changes seem to be context- and species-dependent, lower levels of H3K9me3 are also observed in the spleens of aged rats, which exhibit a concomitant decrease in the levels of the H3K9 methyltransferase SUV39H1 [118]. Indeed, double knockout of *Suv39h1* and *Suv39h2* recapitulates many defects of immunosenescence in mice, including thymic involution, decreased lymphocyte production, higher memory/naïve cell ratio and more HSC priming toward the myeloid lineage. Moreover, regulation of H3K9me3 by SUV39H1 has been shown to determine fate decisions in naïve CD8^+^ T cells. In the absence of SUV39H1, CD8^+^ T cells are unable to repress memory transcriptional programs and therefore impair the capacity to acquire effector functions. Instead, a higher percentage of CD8+ T cells develop into memory T cells, which results in sustained survival and increased long-term memory. Therefore, SUV39H1-mediated H3K9me3 seems to be important for silencing memory programs upon activation in T cells, potentially influencing the decrease in the naïve repertoire observed during aging [120].

Of the many roles that sirtuins play in maintaining heterochromatin in non-immune cells, SUV39H1 is one key target of SIRT6 and SIRT1, suggesting that they may also be important for regulating H3K9me3 in immune cells. SIRT6 mediates the monoubiquitination of SUV39H1, which prevents its binding to chromatin and thereby its H3K9 methylation activity [68]. Conversely, SIRT1 directly regulates SUV39H1 function by deacetylation. In the absence of SIRT1, SUV39H1 activity is dramatically impaired, resulting in loss of H3K9Ac and heterochomatin protein 1 α (HP1α) foci and, in turn, heterochromatin destabilization [11].

Finally, SIRT6 is also involved in immunosenescence and inflammaging since it regulates T cell inflammatory responses. Seminal studies of the role of SIRT6 in aging showed that *Sirt6^-/-^* mice displayed a severe progeroid phenotype involving profound lymphopenia and died within the first month of life. However, *Sirt6^-/-^* lymphocytes normally developed in competitive transplantation assays, indicating a cell-extrinsic phenotype [25]. A subsequent study reported massive multiorgan inflammation in *Sirt6^-/-^* mice, most prominently in their livers. Histological analysis indicated strong infiltration of CD3^+^ T cells and, to a lesser extent, of F4/80^+^ macrophages [121]. In this study, targeted deletion of Sirt6 in T cells or in the myeloid lineage, but not in hepatocytes, recapitulated the inflammatory and fibrotic phenotype in the liver, indicating that SIRT6 regulates inflammation in an immune-cell-autonomous manner. While SIRT7 has not been explicitly studied in the context of immunosenescence, Huo and coworkers reported that high levels of *SIRT7* expression in breast cancer cells are correlated with poor prognosis, T cell exhaustion, and infiltration of pro-inflammatory M1-type macrophages [122], suggesting that SIRT7 activity may contribute to inflammaging and be detrimental to T cell homeostasis during aging.

## 10. B Cells

B cells are the cornerstone of humoral adaptive immunity. Mature B cells reside mainly in the spleen and lymph nodes and express antigen-specific immunoglobulins in their membranes. Upon infection, antigen-specific B cell clones become activated and differentiate into antibody-secreting plasma cells or long-living memory B cells.

The intrinsic functions of sirtuins in B cell aging remain to be explored, but their extensive participation over the lifespan of various cell types, together with reports linking them with B cell maturation and function, suggests that sirtuin dysregulation could contribute to this process. So far, sirtuins have mainly been linked to immunoglobulin class-switching recombination (CSR), B cell homeostasis, and autoimmunity restraint (Figure 2 and Figure 6B). A reduction in the naïve B cell pool occurs during immunosenescence, in part due to a reduced output of immature B cells from the bone marrow [123].

SIRT7 has been reported to be mechanistically important for Non-Homologous End Joining (NHEJ) Repair, a DNA repair pathway on which immunoglobulin CSR relies. *Sirt7^-/-^* splenic B cells display partial defective in vitro class switching (Figure 6B). Even though it is not known whether SIRT7 protein levels change in aged B cells, reduced SIRT7 expression during aging could be partly responsible for the reduction in the clonal diversity of newly generated B cells in response to the immune challenges that are presented during immunosenescence [18]. SIRT1 is also important for proper class switching (Figure 6B). SIRT1 high expression in resting B cells is suppressed upon exposure to class switching-inducing stimuli. Immunoglobulin CSR is mediated by activation-induced cytidine deaminase (AID), whose expression in mature B cells is epigenetically regulated. Overexpression of SIRT1 or enhanced SIRT1 activity results in decreased AID expression and impaired CSR, while pharmacological (Table 1) and genetic inactivation of SIRT1 boosts AID expression and thereby allows proper immunoglobulin maturation. Indeed, SIRT1 has been shown to deacetylate H3K9Ac/H3K14Ac in the promoter of AID, probably as a prelude to heterochromatin induction in the absence of CSR-inducing stimuli [46]. It remains to be established whether SIRT1 downregulation in aged B cells impairs late B cell differentiation in the elderly and has a significant impact on the reduced ability of an aged immune system to respond to vaccines or new infectious challenges.

Inflammaging and the advent of cellular and humoral autoimmune disorders are typically observed during immunosenescence. Highlighting pathological SIRT1 regulation in mature B cells, systemic lupus erythematosus patients display significantly low *SIRT1* levels, and *SIRT1* is negatively correlated with an increased frequency of CD19^+^ B cells in these patients. SIRT1 activation is protective in a murine model of rheumatoid arthritis, in part by decreasing production of autoantibodies by B cells [124]. In asthma patients, serum SIRT1 levels are positively correlated with IgE levels [125], and anti-SIRT1 autoantibodies are more abundant in inflammatory diseases such as psoriatic arthritis [126]. Finally, SIRT1 has also been found to restrain B cell autoimmunity in mouse models. The liver and kidneys of *Sirt1^-/-^* mice accumulate deposits of immune complexes, and antibodies that react with nuclear antigens were found in the sera of these mice [107].

SIRT1 is also necessary for B cell antigen presentation, as *Sirt1*-deficient B cells express reduced levels of MHC-II and maturation markers, resulting in reduced cross-presentation to CD4^+^ T cells [104]. Counter to this pathological scenario, SIRT1 overexpression in the transformed mouse B lymphocyte BaF3 cell line increases cell proliferation and cytokine production while reducing apoptotic rates [127]. Indeed, human antibody-secreting plasma cells have been shown to express higher *SIRT1* levels than resting B cells, probably reflecting the increased metabolic demand required for the former to be active. Circulating B cells from aged individuals express lower levels of SIRT1 than young people [128]. Together, these data indicate that SIRT1 downregulation with age may contribute to immunosenescence by establishing a propensity to lose B cell tolerance.

## 11. Conclusions and Future Directions

In this review, we have comprehensively summarized our current knowledge about sirtuin functions in the context of immune system aging (Figure 7). Except for SIRT1, which is by far the most widely studied sirtuin in the context of immunosenescence, few studies have explored the participation of other nuclear sirtuins in the aging of the immune system, so our knowledge of the importance and specific roles of sirtuins in this context remains quite fragmentary. Nuclear sirtuins maintain genome integrity through the deployment of epigenetic mechanisms, and it is well-established that epigenetic alterations accompany immunosenescence [53]. Likewise, the transcriptional activity of specific transcription factors is affected during immunosenescence, and some of these act in concert with sirtuins [99,105]. Sirtuins are broadly acknowledged to have an important influence on the lifespan and health-span across species, so we may expect that in the coming years we will witness the identification of specific molecular pathways and mechanisms involved in the activity of sirtuins in immune system aging.

Important questions remain, such as those surrounding the sirtuin expression pattern in aged human immune populations and its possible correlation with the onset of age-related signs such as frailty and age-related maladies. Some evidence indicates that sirtuins are more strongly expressed in the immune system than in other tissues and organs, which suggests that systemic sirtuin targeting may substantially affect immune reactivity. The fact that sirtuins are both pharmacologically and dietarily targetable makes them promising targets for therapeutic intervention in immunosenescence, in particular, and in organismal aging, in general. Sirtuins have the virtue of being pharmacologically activable, which suggests avenues for potentially counteracting the diminished activity of sirtuins during aging and for improving particular aspects of immunosenescence. Despite this optimism, caution is needed, as the ubiquity of sirtuins may create challenges in the delivery of tissue-specific treatments. Further research is expected to provide a more precise and integrative conceptual framework of sirtuin functions in immune and organismal cell aging, which may help to evaluate their potential targets for immunosenescence therapy.

## Figures and Tables

**Figure 1 genes-12-01856-f001:**
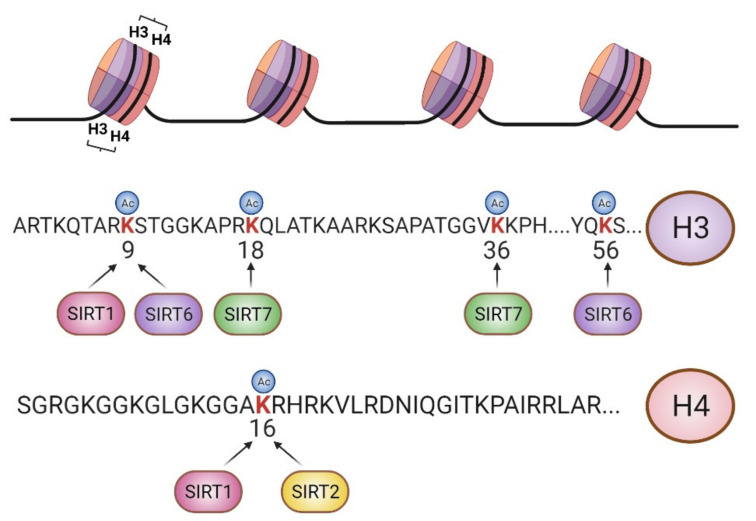
Scheme of major epigenetic targets of sirtuins. Nuclear sirtuins target specific lysine residues of histones H3 and H4 to develop chromatin-associated functions [9]. SIRT1 deacetylates H3K9ac and H4K16ac, and SIRT2 deacetylates H4K16ac during the G2/M transition. SIRT6 deacetylates H3K56ac, H3K18ac and H3K9ac, and SIRT7 deacetylation activity is directed towards H3K18ac and H3K36ac. Figure created with BioRender.com.

**Figure 2 genes-12-01856-f002:**
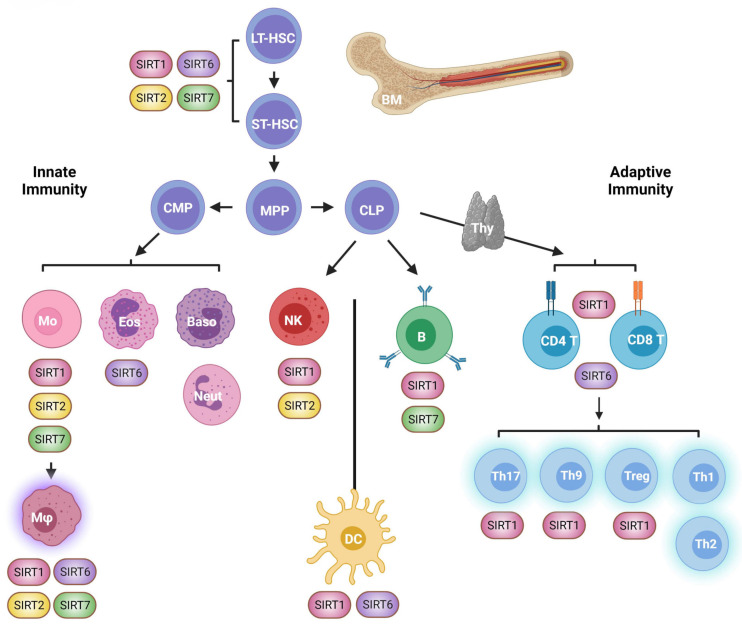
Overview of immune system ontogeny and its relationship with nuclear sirtuin activity. Hematopoiesis starts in the bone marrow through sequential differentiation of hematopoietic stem cells (HSC) into different immune cell progenitors (upper part). Innate immune cells originate from a common myeloid progenitor (MCP) and involve a variety of immune cell types including monocytes (Mo), macrophages (Mφ), eosinophils (Eos), basophils (Baso), and neutrophils (Neu) (left lower part). Common lymphoid progenitors (CLP) give rise to B cells in the bone marrow and T cells in the thymus (thy) (right lower part). Natural killer (NK) cells, despite having a lymphoid origin, play important roles in innate immunity against tumors and viral-infected cells. Dendritic cells (DC) have a lymphoid and myeloid origin (not shown in this figure) and sit at the interphase of innate and adaptive immunity. Sirtuins icons denote sirtuin cell type-specific roles. ST-HSC: short-term HSC; LT-HSC: gong-term; MPP: Multipotent progenitor. Figure created with BioRender.com.

**Figure 3 genes-12-01856-f003:**
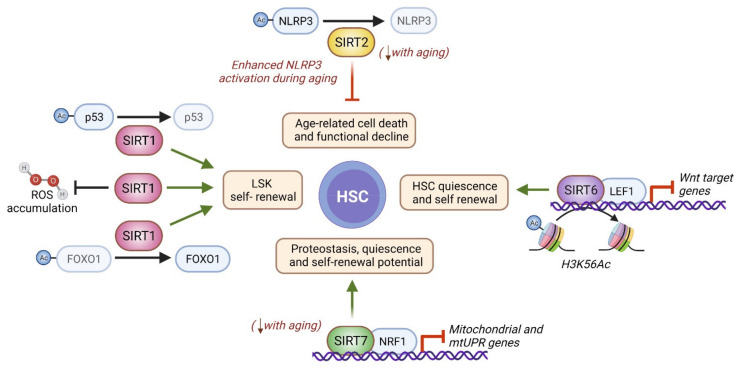
Molecular basis of nuclear sirtuin functions during HSC aging. In vitro cultured LSK cells from Sirt1^-/-^ mice show lower self-renewal capacity as a consequence of a mechanism involving FOXO suppression, p53 activation, and ROS accumulation [35]. In HSCs from young mice, SIRT2 inactivates the NLRP3 inflammasome by deacetylation to mitigate cell death. During aging, SIRT2 becomes downregulated in HSC, which provokes enhanced NLRP3 activation and age-related HSC functional decline [36]. SIRT6 is recruited by the Wnt transcription factor LEF to the promoters of Wnt target genes, where it deacetylates H3K56Ac to suppress their expression. In this way, SIRT6 contributes to the maintenance of HSC quiescence and self-renewal potential [37]. Finally, SIRT7 is recruited to mitochondrial- and mtUPR-related genes in a NRF1-dependent manner in HSCs to repress their expression and ensure mitochondrial proteostasis and HSC quiescence and self-renewal potential [38]. Faded lines indicate age-related loss of function, and comments in red indicate age-related changes. Figure created with BioRender.com.

**Figure 4 genes-12-01856-f004:**
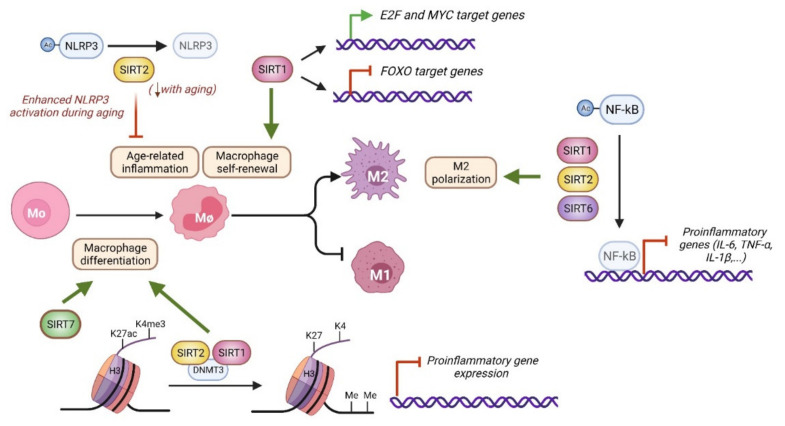
Molecular pathways related to nuclear sirtuin activity in macrophages. Macrophages differentiate from bone marrow cells or self-maintain through local proliferation. After antigen challenge, macrophages polarize into M1 and M2 subtypes. SIRT1 and SIRT2 prevent the premature expression of proinflammatory genes during macrophage differentiation by controlling chromatin structure, including DNA methylation and H3K27ac and H3K4me3 levels [47]. SIRT1 also regulates macrophage self-renewal through MYC, E2F and FOXO dependent pathways [42]. SIRT2 limits age dependent inflammation through NLRP3 deacetylation [63]. SIRT7 stimulates monocyte to macrophage differentiation, but the underlying molecular mechanism remains unclear [64]. SIRT1, SIRT2, and SIRT6 promote M2 polarization by limiting NF-kB activity and reducing the expression of proinflammatory cytokines [57,58,65].

**Figure 5 genes-12-01856-f005:**
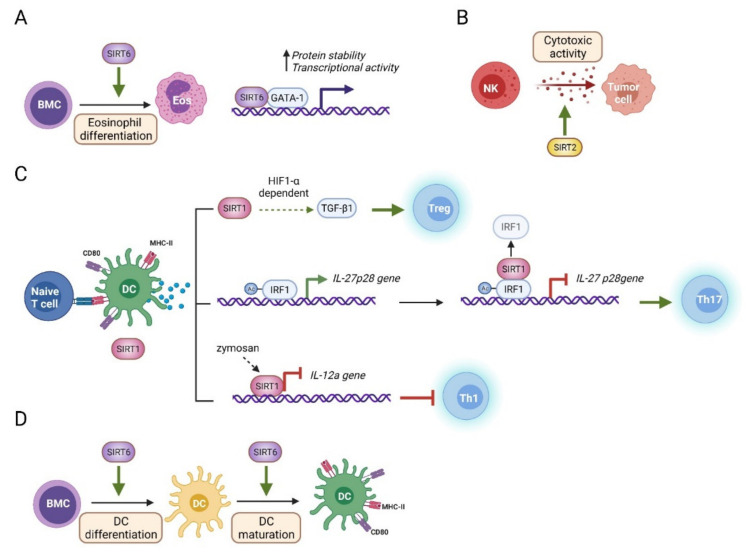
Molecular pathways related to nuclear sirtuin activity in innate immune cells. (**A**) SIRT6 promotes eosinophil differentiation, possibly through GATA-1 stabilization and activation [75]. (**B**) SIRT2 enhances NK cell-mediated cytotoxicity versus hepatocellular carcinoma cells, but the underlying molecular mechanism remains mostly unknown [79]. (**C**) In DCs, SIRT1 regulates cytokine expression with important consequences for subsequent Th differentiation [80,81,82]. SIRT1 promotes Treg differentiation through TGF-β1 production in a HIF1-α dependent manner. SIRT1 also promotes Th17 differentiation through IRF1 deacetylation, thereby limiting its binding to the il-27p28 promoter and silencing its expression. Moreover, in response to zymosan, SIRT1 gets recruited to the il-12a gene promoter to repress its expression and limit Th1 differentiation. (**D**) SIRT6 is required for both DC differentiation and maturation, but the molecular mechanisms involved have not been explored [48].

**Figure 7 genes-12-01856-f007:**
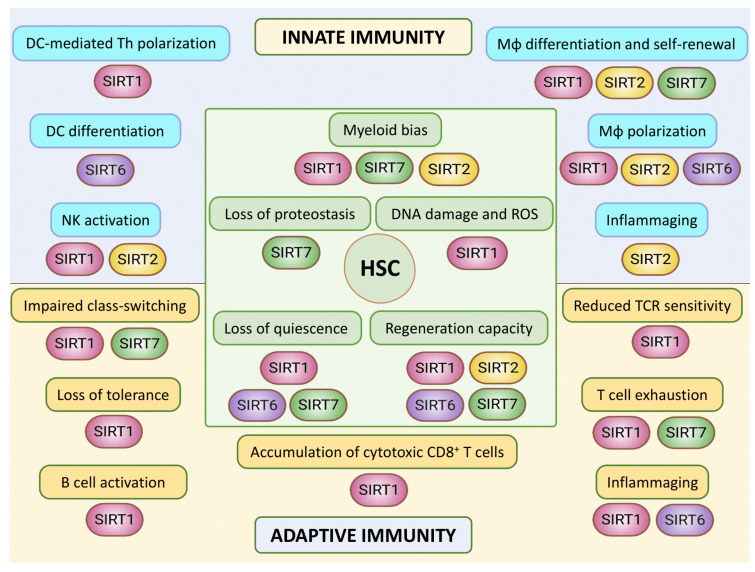
Hallmarks of immune aging and their connections to nuclear sirtuins. In the central part, different features associated with the aging of hematopoietic stem cells (HSCs) and their relationship with sirtuin activity are depicted. Nuclear Sirtuins are important to prevent several aspects of HSC aging, including DNA damage, myeloid bias and loss of proteostasis and quiescence, as well as maintenance of HSC regeneration capacity [31,35,36,37,39]. In the upper part, aging-associated features of innate immune cell types are summarized. SIRT1 and SIRT6 are important for DC-mediated Th polarization and DC differentiation, respectively [48,80,81,82]. SIRT1, SIRT2, SIRT6 and SIRT7 also play relevant roles in macrophage (Mφ) biology, and SIRT2 has been linked to inflammaging [47,57,58,65]. In NK cells, SIRT1 and SIRT2 regulate their cytotoxic functions in the contexts of aging and anticancer response, respectively [79,84]. In the lower part, characteristics of the aging of the adaptive immune cell compartment are shown. SIRT1, SIRT6 and SIRT7 are important for several aspects of T cell biology and aging [40,95,100,109,111,112]. SIRT1 is further involved in B cell activation and immunoglobulin gene class-switching, where SIRT7 also plays a role [16,46,104,127,128].

## Data Availability

Not applicable.

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
