# Peer review of "Nuclear Sirtuins and the Aging of the Immune System"

_genes, 2021, doi:10.3390/genes12121856_

Round 1

Reviewer 1 Report

Review is very well researched, and the figures complement the text well. The discussion spent on innate as well as adaptive immunity and the implicated sirtuins adds to the body of literature. 

Very minor suggestions:

Figure 2- it may be helpful to flip innate immunity to left and adaptive to right to reflect order of events and discussion of results.

Figure 3- flip innate and adaptive immunity order and discussion of results.

Author Response

Reviewer 1 

We thank the reviewer for his/her comments on our manuscript. In addition to include the reviewer's specific comments, we have also checked the full-text manuscript and corrected some grammatical errors. 

Q1: Figure 2- it may be helpful to flip innate immunity to left and adaptive to right to reflect the order of events and discussion of results. 

A1: We have changed the relative order of innate and adaptive immunity in figure 2. In Figure 2, innate immunity is now on the left and adaptive immunity on the right.  Figure legend has been also changed accordingly. 

Q2. Figure 3- flip innate and adaptive immunity order and discussion of results. 

We have also changed the relative order of innate and adaptive immunity in new Figure 7 (previous figure 3). In new Figure 7, innate immunity is in the upper part, and adaptive immunity is in the lower part. Figure legend has been also changed accordingly. 

Reviewer 2 Report

I very much enjoyed reading this review and I think it will be useful to the field of immunosenescence. 

Minor comments:

A table summarizing the effect of different nuclear SIRT agonists/antagonists on the aging of the discussed immune cell populations would be useful.

Line 96 - "Sirtuins are strongly expressed". Can you reword this to highly expressed, or abundant, etc.

Line 98 - "Innate and immune responses". Should be "innate and adaptive immune responses" I assume?

Line 327 - "SIRT7 expression, while and SIRT7 expression". A typo, please correct.

Line 373 - " and allergic inflammations" should be inflammation (singular)

Author Response

We thank the reviewer for his/her comments and helpful suggestions for improving the manuscript.   

Q1: A table summarizing the effect of different nuclear SIRT agonists/antagonists on the aging of the discussed immune cell populations would be useful. 

A1: We thank the reviewer for this helpful suggestion. The role of sirtuin modulators in the context of immune aging has been little explored. Nevertheless, we have included a table summarizing the role of Sirtuin modulators and their effects on immune populations. 

Q2: Line 96 - "Sirtuins are strongly expressed". Can you reword this to highly expressed, or abundant, etc.  

Line 98 - "Innate and immune responses". Should be "innate and adaptive immune responses" I assume? 

A2: We thank the reviewer for pointing out these two mistakes. We have changed the word “strongly” to “highly”. In addition, we have corrected the sentence “innate and immune responses” to “innate and adaptive immune responses”. Now the full sentence reads as: 

“Sirtuins are highly expressed in immune cells and play multiple roles in cytokine production, inflammation, and the development of innate and adaptive responses” 

Q3. Line 327 - "SIRT7 expression, while and SIRT7 expression". A typo, please correct. 

A3. We thank the reviewer for pointing out this mistake. We have removed the extra word “and”. 

Q4. Line 373 - " and allergic inflammations" should be inflammation (singular) 

A4. We thank the reviewer for pointing out this mistake. We corrected the word “inflammations” to “inflammation”. 

Reviewer 3 Report

The review article by Andrés Gámez-García and Berta N. Vazquez focuses on the role of nuclear sirtuins in the immune processes and their potential role in in the aging of the latter. The article provides a comprehensive overview of these processes, and the authors do a good job in summarizing the data currently available in the literature.

However, the lack of structure of the information presented and the lack of figures showing the basis of the molecular mechanisms occurring between each sirtuins and the regulators/interactors prevents the reader to understand the complex view that the authors aim to present.

In order to improve the structure and the logic of the information discussed in the manuscript, the following points should be addressed:

  1. The title of the review article “Sirtuins and the aging of the immune system” is misleading, because the review article does not present the role of ‘all’ sirtuins but focus on the role of nuclear sirtuins. The title should be amended to reflect the content of the article (for example, “Nuclear sirtuins and the aging of the immune system”). Alternatively, the authors may further expand the review article to include the role and connections of ‘all’ sirtuins in the aging of the immune system.
  2. The authors refer to the role of nuclear sirtuins at the organismal level only in the last paragraph of the Introduction (just about six lines of text). The most up-to-date systematic review is available (Maissan et al, Biology (Basel), 2021) about the role of sirtuins in all mammalian tissues, and it may be referred to as it represents a useful starting point to briefly introduce the role of these molecules at the systemic/organismal level, and then to highlight their specific role in the nucleus in the known tissues/organs. The authors mention only the brain, but the evidence of SIRT1, SIRT6 and SIRT7 in the nucleus of other tissues may be mentioned.
  3. All main sections (2 through 10) appear to be clear in term if which nuclear sirtuin does what in the various context. However, in each section, the regulatory mechanisms through which individual nuclear sirtuins are provided without making a clear connection among the information presented. In each section, the inclusion of a scheme/drawing would be helpful for the reader to follow the concepts presented in a logic manner.
  4. Figure 2 and Figure 3 nicely show the main involvement of nuclear sirtuins in the adaptive and innate immunity. However, each statement indicating an experimental fact is currently not appropriately referenced and should be inserted.
  5. The specific roles sirtuins have in each of the processes indicated in Figure 2 and Figure 3 are not discussed in full in the text. Because this is a review article, all details referring to the main aspects aimed for the piece should be provided. Specifically, each process mentioned in the figures, where a specific sirtuin or multiple sirtuins play a role, should be detailed in the text (specifically in section 2 and section 3). The information should be not listed as a series of facts, but the facts should be connected in a logic manner to one another. This way, the reader will be able to understand the details of the sirtuins involvement in the immune pathways at a system-level.

Author Response

We thank the reviewer for his/her comments and helpful suggestions for improving the manuscript.   

Q1: The title of the review article “Sirtuins and the aging of the immune system” is misleading, because the review article does not present the role of ‘all’ sirtuins but focus on the role of nuclear sirtuins. The title should be amended to reflect the content of the article (for example, “Nuclear sirtuins and the aging of the immune system”). Alternatively, the authors may further expand the review article to include the role and connections of ‘all’ sirtuins in the aging of the immune system. 

A1: We thank the reviewer for bringing this up. We agree with her/him that it is a good idea to change the tile of the manuscript. In the revised version, the tiles now read as “Nuclear sirtuins and the aging of the immune system”.  

Q2: The authors refer to the role of nuclear sirtuins at the organismal level only in the last paragraph of the Introduction (just about six lines of text). The most up-to-date systematic review is available (Maissan et al, Biology (Basel), 2021) about the role of sirtuins in all mammalian tissues, and it may be referred to as it represents a useful starting point to briefly introduce the role of these molecules at the systemic/organismal level, and then to highlight their specific role in the nucleus in the known tissues/organs. The authors mention only the brain, but the evidence of SIRT1, SIRT6 and SIRT7 in the nucleus of other tissues may be mentioned. 

A2: The goal of the mentioned paragraph is to provide an overview of the different nuclear sirtuin animal models known to correlate with either longevity or premature aging phenotypes. Nevertheless, we thank the reviewer for redirecting us to the excellent paper written by Maissan P and colleagues, which has been included at the beginning of the paragraph and it reads as  “Sirtuins are ubiquitously expressed proteins with important roles in numerous tissues (Maissan et al, Biology, 2021).” 

Q3: All main sections (2 through 10) appear to be clear in term if which nuclear sirtuin does what in the various context. However, in each section, the regulatory mechanisms through which individual nuclear sirtuins are provided without making a clear connection among the information presented. In each section, the inclusion of a scheme/drawing would be helpful for the reader to follow the concepts presented in a logic manner. 

A3: We thank the reviewer for making this suggestion. We have made four extra figures to reflect in more detail the molecular mechanisms by which nuclear Sirtuins regulate immune cell function. Figure 3 is dedicated to HSC, Figure 4 to macrophages, Figure 5 to other innate immune cells (Eosinophils, NK cells and dendritic cells) and Figure 6 to adaptive immune cells.  

Q4: Figure 2 and Figure 3 nicely show the main involvement of nuclear sirtuins in the adaptive and innate immunity. However, each statement indicating an experimental fact is currently not appropriately referenced and should be inserted. 

A4: We thank the reviewer for pointing out this mistake. We have included citations in the legends of all figures. 

Q5: The specific roles sirtuins have in each of the processes indicated in Figure 2 and Figure 3 are not discussed in full in the text. Because this is a review article, all details referring to the main aspects aimed for the piece should be provided. Specifically, each process mentioned in the figures, where a specific sirtuin or multiple sirtuins play a role, should be detailed in the text (specifically in section 2 and section 3). The information should be not listed as a series of facts, but the facts should be connected in a logic manner to one another. This way, the reader will be able to understand the details of the sirtuins involvement in the immune pathways at a system-level. 

A5. We thank the reviewer for expressing his/her concerns on whether the information found in the figures is properly explained in the text.  To better support the molecular event explained in the main text, we have made four extra figures (Fig.4-6) that we hope will enhance clarity and readability. 

Round 2

Reviewer 3 Report

I thank the authors for addressing in full the comments that I have raised in the previous version of the manuscript. Particularly, the authors have made a great, appreciable effort to include additional figures that visualize the (molecular) mechanisms described in the text, and to indicate in the figure legends the relevant literature in support to the figures.

As a minor note, Figure 5D may be not cited in the text before Figure 5, and Figure 7 may be not cited in the text before Figure 6A.

Author Response

We thank the reviewer for his/her careful revision of our manuscript and we appreciate his/her comments to improve manuscript quality. We have corrected the order of figure citations in the main text as suggested by the reviewer.